# Powders Based on $Ca_2P_2O_7$-$CaCO_3$-$H_2O$ System as Model Objects for the Development of Bioceramics

Kristina Peranidze [1,*], Tatiana V. Safronova [1,2], Yaroslav Filippov [1,3], Gilyana Kazakova [1,4], Tatiana Shatalova [1,2] and Julietta V. Rau [5,6]

1   Department of Materials Science, Lomonosov Moscow State University, Leninskie Gory 1, Building 73, 119991 Moscow, Russia
2   Department of Chemistry, Lomonosov Moscow State University, Building 3, Leninskie Gory, 1, 119991 Moscow, Russia
3   Research Institute of Mechanics, Lomonosov Moscow State University, Michurinsky 1, 119192 Moscow, Russia
4   Institute for Regenerative Medicine, Sechenov First Moscow State Medical University, Trubetskaya 8, Build. 2, 119991 Moscow, Russia
5   Istituto di Struttura della Materia, Consiglio Nazionale delle Ricerche (ISM-CNR), Via Del Fosso del Cavaliere 100, 00133 Rome, Italy
6   Department of Analytical, Physical and Colloid Chemistry, Institute of Pharmacy, Sechenov First Moscow State Medical University, Trubetskaya 8, Building 2, 119991 Moscow, Russia
*   Correspondence: perika5@mail.ru; Tel.: +7-9057967636

**Abstract:** Nanoscale powders of hydrated $Ca_2P_2O_7$, $CaCO_3$, and a product of mixed-anionic composition containing $P_2O_7^{4-}$ and $CO_3^{2-}$ anions were synthesized from aqueous solutions of $Ca(CH_3COO)_2$, pyrophosphoric acid ($H_4P_2O_7$), and/or $(NH_4)_2CO_3$. Pyrophosphoric acid was previously obtained on the basis of the ion exchange process from $Na_4P_2O_7$ solution and $H^+$-cationite resin for further introduction into the reactions as an anionic precursor. The phase composition of powders after the syntheses was represented by bioresorbable phases of X-ray amorphous hydrated $Ca_2P_2O_7$ phase, calcite and vaterite polymorphs of $CaCO_3$. Based on synthesized powders, simple cylindrical constructions were prepared via mechanical pressing and fired in the temperature range of 600–800 °C. Surface morphology observation showed the presence of bimodal porosity with pore sizes up to 200 nm and 2 μm, which is likely to ensure tight particle packing and roughness of the sample surface required for the differentiation of osteogenic cells. Thus, the prepared ceramic samples can be further examined as model objects for bone tissue repair.

**Keywords:** calcium pyrophosphate; powders; bioceramics; calcium carbonate; bone tissue repair

## 1. Introduction

To date, various calcium phosphates have been investigated in biomedical engineering as compounds for the production of biocompatible, biodegradable ceramic constructions intended to replace damaged areas of bone tissue. Despite the fact that significant research interest is focused on the synthesis of hydroxyapatite ($Ca_{10}(PO_4)_6(OH)_2$, Ca/P = 1.67) [1,2] and calcium orthophosphate ($Ca_3(PO_4)_2$, Ca/P = 1.5) [3,4] phases occurring naturally in bone tissue, special attention is paid to calcium phosphates with a Ca/P molar ratio different from the that of natural inorganic bone constituents. For instance, calcium metaphosphate [5], brushite [6,7], calcium polyphosphate [8], calcium pyrophosphate ($Ca_2P_2O_7$, CPP) [9,10], as well as their biphasic composites [11] and ion-substituted analogs [12] are widely described in the literature. It is also noteworthy that such calcium phosphate materials can act not only as independent ceramic bases for the fabrication of bone implants but also as bioactive ceramic inclusions in bioglass- [13] or polymer-based systems that are currently being studied as promising materials for cell culturing and drug delivery in the field of bioscaffolding [14,15]. The role of calcium pyrophosphate in developing bioceramics is quite ambiguous. A number of studies admit that CPP ceramics can be associated

with the abundant calcium pyrophosphate dihydrate ($Ca_2P_2O_7 \cdot 2H_2O$) deposition in the meniscus or intervertebral disks [16], which, as a rule, leads to degenerative arthritis [17]. However, notwithstanding this assumption, the authors suppose that the appearance of arthritis may be caused by specific pathological processes of individual organisms. However, the relatively good tendency to dissolve in aqueous media (resorption) without rapid formation of insoluble hydroxyapatite phase, as well as moderate biocompatibility caused, among other parameters, by slightly neutral pH when immersed in water with tissues and cellular fluids, allow scientists to consider CPP-based ceramics promising material for bone tissue repair [18–20].

The key issue when fabricating CPP-based ceramic constructions is related to the obstructed mass transfer that takes place due to weak diffusion of large, multiply charged pyrophosphate anions. Although high temperatures and lengthy sintering can slightly accelerate the process, the mechanical properties of bioceramics are likely to remain poor due to the low density of the material [21]. Thus, solid-phase sintering does not give the desired effect and is usually replaced with liquid-phase sintering using such additives as sodium phosphate [22], sodium carbonate [23], calcium chloride [24], potassium chloride [25], calcium polyphosphates [26], etc. In [27], the formation of low-melting eutectics with CPP is described. Another way to facilitate mass transfer can be associated with co-precipitation of CPP together with additional components from an aqueous solution during the synthesis of the initial powder. Such a method leads to the formation of a two-phase powder as a target product and allows it not to be cleaned from by-products containing $Na^+$, $K^+$, $Cl^-$, $NO_3^-$ or other ions.

Carbonate anion is considered the main source of hydroxyapatite crystal lattice deformation, specifically, $CO_3^{2-}$ groups taking B-position in the structure of hydroxyapatite participate in dissolution–precipitation processes in the presence of extracellular body fluids. The significant role of $CO_3^{2-}$ groups in biochemical interactions between blood plasma and tissues was emphasized in [28]. A number of studies [29,30] on bioceramics development report that the addition of calcium carbonate ($CaCO_3$) to a calcium phosphate-containing system was carried out in order to regulate the resorption and improve the mechanical properties of the powders. Thus, the development of ceramic materials based on nano-/macroscale powders in the calcium phosphate–calcium carbonate system seems potentially promising.

Since the development of pyrophosphate–carbonate ceramics as a potential material for the elimination of bone tissue defects is poorly presented in the literature to date, it is proposed in this study to consider calcium carbonate specifically in combination with calcium pyrophosphate. The use of CPP as a phosphate component in parallel with the studies in the orthophosphate–carbonate system is justified by the intention to compare both the properties of the final product and the conditions of material preparation that includes a specific synthesis method, disaggregation of powder product, and molded material behavior during heat treatment necessary for the ceramics production. Additionally, the issues while working with $P_2O_7^{4-}$ anion described above, along with difficulty in sintering carbonates, which are liable to decompose at low temperatures, make such a concept a challenging, not largely reported task that may pave the way to the development of composite carbonate-containing ceramics intended for biomedical purposes.

The prospects of the development of carbonate-containing ceramics for the treatment of congenital or acquired bone tissue injuries, provided that the typical disadvantages of calcium carbonate thermal treatment are eliminated, can play a positive role in the biochemical processes of tissue healing, taking into account the above-described properties of $CO_3^{2-}$ anion. The use of 3D biomaterials based on such composite ceramics as individual bone implants or complex constructions in combination with bioactive organic compounds accelerating tissue growth is considered a substantial breakthrough in regenerative medicine. Another potential field of application of composite ceramics based on $Ca_2P_2O_7$-$CaCO_3$-$H_2O$ systems is associated with the development of dental materials. The recent research [31] provides a discussion on the functional material Activa$^{TM}$ bioactive

restorative composite that combines the benefits of composites and glass ionomers and can be applied to restore primary molars.

Therefore, the current research is devoted to the fabrication of model ceramic materials obtained based on CPP and calcium carbonate powders and the study of the basic properties of the samples, including phase composition, morphology, thermal and mechanical stability.

## 2. Materials and Methods

### 2.1. Synthesis of Powders

Powders of calcium acetate $Ca(CH_3COO)_2$ and ammonium carbonate $((NH_4)_2CO_3)$ were used for the synthesis of $CaCO_3$. The chemical reaction occurs in accordance with Equation (1):

$$Ca(CH_3COO)_2 + (NH_4)_2CO_3 = CaCO_3\downarrow + 2CH_3COONH_4 \qquad (1)$$

For the synthesis of CPP powder and a product of mixed-anionic composition containing coprecipitated $P_2O_7^{4-}$ and $CO_3^{2-}$ anions, it was necessary to obtain pyrophosphoric acid beforehand. Pyrophosphoric acid was prepared using an ion exchange from a solution of sodium pyrophosphate $(Na_4P_2O_7)$ in accordance with the method reported in [32]. The ion exchange process was carried out using the ion-exchange resin KU-2-8 ($H^+$ cationite). However, the ion exchange process is characterized by a certain stability constant and, therefore, does not proceed completely. Thus, it is possible to obtain a solution of pyrophosphoric acid containing a certain amount of sodium ions. The resin was taken in a fourfold excess compared to the mass of sodium pyrophosphate.

The ion-exchange resin was prepared for use by holding it for 30 min in distilled water. After washing, the resin was separated from the liquid by filtration on a Buchner funnel. The operation was repeated twice. Then, a fourfold excess of resin was added to a solution of sodium pyrophosphate. The synthesis was carried out on a magnetic stirrer for 45 min. At the same time, a transparent solution containing hydrogen ions and pyrophosphate ions, pH = 1, was formed. The ion exchange process can be reflected using Equation (2):

$$Na_4P_2O_7 \text{ (aq)} + H^+ \text{ (resin)} = H^+ \text{ (solution)} + P_2O_7^{4-} \text{ (solution)} + 4Na^+ \text{ (resin)} \qquad (2)$$

The resulting transparent solution ($H_4P_2O_7$ containing sodium ions) was separated by filtration on a Buchner funnel and neutralized with an ammonia solution to pH = 7. Then a solution of calcium acetate was added. Equations (3) and (4) reflect the synthesis of CPP powder and powder of mixed anionic solution, respectively.

$$H_4P_2O_7 + 2Ca(CH_3COO)_2 + 4NH_3 \cdot H_2O + (x-4)H_2O = Ca_2P_2O_7 \cdot xH_2O\downarrow + 4CH_3COONH_4 \qquad (3)$$

$$H_4P_2O_7 + 3Ca(CH_3COO)_2 + (NH_4)_2CO_3 + 4NH_3 \cdot H_2O + (x-4)H_2O = Ca_2P_2O_7 \cdot xH_2O\downarrow + CaCO_3\downarrow + 6CH_3COONH_4 \qquad (4)$$

The key parameters of the syntheses in the calcium pyrophosphate–calcium carbonate system are shown in Table 1.

**Table 1.** Syntheses of powder products in $Ca_2P_2O_7$-$CaCO_3$-$H_2O$ system.

| Target Product | Labeling | Concentrations | | |
|---|---|---|---|---|
| | | $(NH_4)_2CO_3$ | $(NH_4)_2P_2O_7$ Previously Obtained | $(CH_3COO)_2Ca$ |
| $CaCO_3$ | CC | 0.5 M | - | 0.5 M |
| $Ca_2P_2O_7 \cdot xH_2O/CaCO_3$ | CPP/CC | 0.125 M | 0.125 M | 0.5 M |
| $Ca_2P_2O_7 \cdot xH_2O$ | CPP | - | 0.25 M | 0.5 M |

After drying in a thin layer, each of the precipitated samples was placed in a special container together with a calculated mass of grinding $ZrO_2$-spheres ($m_{powder}$:$m_{spheres}$ = 1:5)

and then ground in a planetary ball mill under acetone for 20 min with subsequent evaporation of the acetone and sieving through a polyester sieve with a mesh size of 200 μm. To compare the shrinkage of ceramic samples powders of hydrated CPP and CPP/CC were ground in the presence of NaCl—$(NaPO_3)_6$ additive taken in an amount of 10 wt %.

## 2.2. Preparation of Ceramic Samples

The samples were pressed into the form of simple cylinders with a mass range of 0.30–0.34 g using steel die with diametre 12 mm. The average diameter and height of the samples after firing were 11.1 mm and 2.1 mm. Pressing was carried out using a mechanical press at a pressure of 100 MPa.

Powders compressed into tablets were fired at 600, 700, and 800 °C. Heating in the furnace was carried out at a rate of 5 °C/min. The holding time at these temperatures was 2 h. The firing of the samples was carried out in order to study the effect of high temperatures on the initial composition, as well as to determine mass losses and outline the thermal behavior of the materials in the presented temperature range. The shrinkage density of the samples after firing was calculated as well, and the role of the sintering additive of eutectic composition was revealed.

## 2.3. Characterization Methods

The phase composition of the obtained samples was investigated on the basis of powders or ceramic chips by X-ray diffraction method (XRD) (Cu Ka radiation, 2θ 2–70°, diffractometer with rotating anode Rigaku D/Max-2500, Japan). Qualitative analysis of the phases was performed using the ICDD PDF2 database. An external standard method (corundum number method) was applied to calculate the approximate mass content of crystalline modifications. Major peak intensities and reference intensity ratio (RIR) were taken using the database of the WINXPOW program.

The microstructure was studied by means of scanning electron microscopy (SEM) using a LEO SUPRA 50VP electron microscope (Carl Zeiss, Germany; field emission source) in the secondary electron mode (SE2 detector) at a working voltage of 5 kV. Before conducting the survey of the samples, a chromium layer (up to 10 nm) was sprayed on the powders or ceramic chips under study to ensure their conductivity.

The behavior of the samples during heat treatment in the temperature range of 25–1000 °C at a heating rate of 10 °C/min was described by synchronous thermal analysis (TG—thermogravimetry) with mass spectrometric detection using thermal analyzer Netzsch STA-409 PC Luxx (Germany) equipped with quadrupole mass spectrometer QMS 403C Aëolos (Germany). Mass spectra were recorded for mass numbers 18 ($H_2O$), 17 ($OH^-/NH_3$), 30 (NO), 46 ($NO_2$), and 44 ($CO_2$).

## 3. Results and Discussions

As a result of $CaCO_3$ synthesis from $Ca(CH_3COO)_2$ and $(NH_4)_2CO_3$, the obtained product included calcite and vaterite crystalline modifications, both of which are found in shells, bones, and corals, which, along with aragonite-type of $CaCO_3$ have been widely used as bone substitutes in the clinic [33]. The quantitative analysis by the external standard method showed a predominance of metastable vaterite phase in the amount of 64 wt %, which may be due to a short duration of precipitate aging, so the initial phase synthesized from aqueous solutions did not completely turn into a stable calcite phase. According to XRD spectra, X-ray amorphous powders of CPP and CPP/CC samples were synthesized from aqueous solutions containing pyrophosphoric acid (Figure 1). As expected, the samples after the syntheses possess a strongly amorphous structure and do not have pronounced peaks on diffractograms. With a significant resolution of the XRD data of CPP powder, a weak characteristic peak can be identified on the diffraction pattern at 28°. The inability to identify coprecipitated phases of CPP/CC product allows us to assume a high uniformity of components distribution in the polycrystalline powder. Such data are generally consistent with the data obtained in an earlier study [29]. Overall, the estimated

compositions of the initial CPP and CPP/CC samples can be determined based on XRD data of the samples after firing and thermal analysis with mass spectrometric detection. In particular, the greatest interest is directed to the study of synergetic effect between the phases in CPP/CC sample manifesting in the morphology evolution.

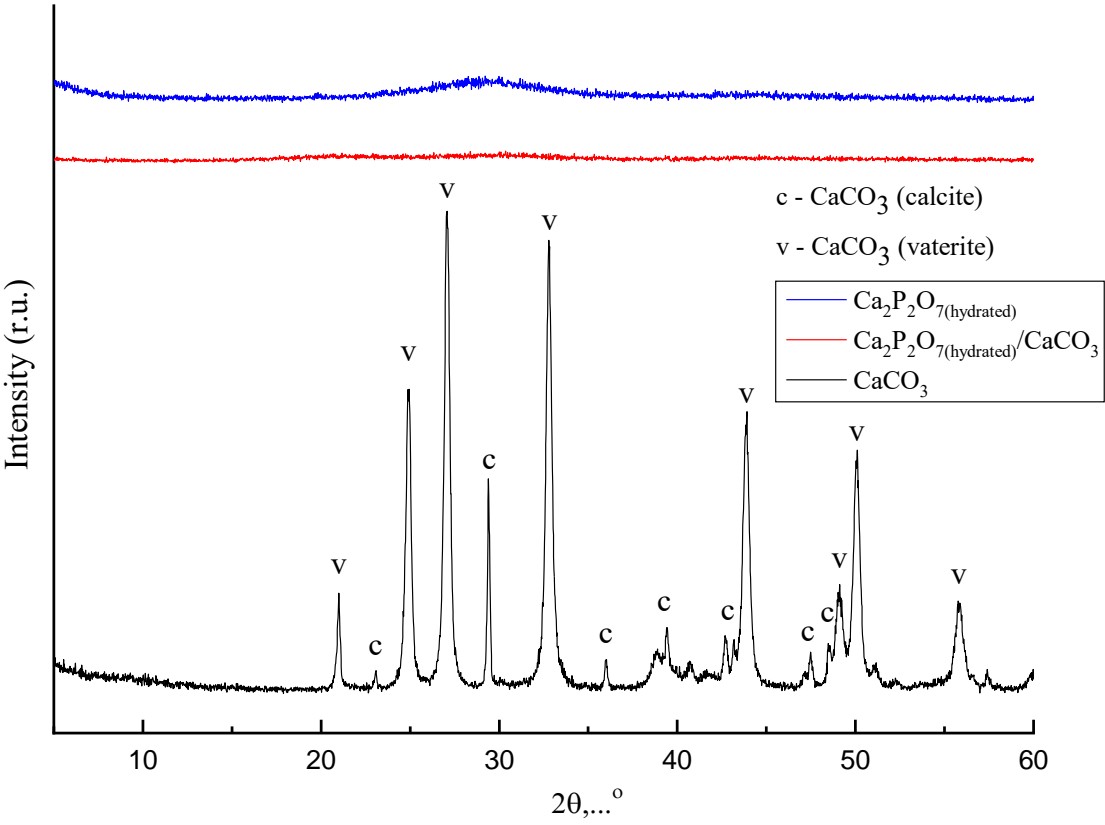

**Figure 1.** XRD data for powders synthesized from calcium acetate and ammonium pyrophosphate (blue diffractogram); calcium acetate and a mixed-anionic solution containing pyrophosphate and carbonate ions (red diffractogram); calcium acetate and ammonium carbonate (black diffractogram).

The XRD data of the samples after firing in the temperature range of 600–800 °C shows that $CaCO_3$ decomposition begins at temperatures > 700 °C with the formation of CaO, which, when interacting with water vapor, forms portlantide $Ca(OH)_2$. The presence of this component in ceramic biomaterials is obviously undesirable since contact with aqueous media leads to an increase in pH level up to 12 (strongly alkaline reaction). When firing a CPP sample, the formation of β-calcium pyrophosphate (β-$Ca_2P_2O_7$) occurs. The firing of the CPP/CC sample prepared on the basis of powder synthesized from a solution of mixed-anionic composition ultimately leads to the formation of the β-$Ca_3(PO_4)_2$ phase as the dominant phase. Figure 2 reflects the phase evolution in the CPP/CC sample from X-ray amorphous powder synthesized from a mixed-anionic solution containing $P_2O_7^{4-}$ and $CO_3^{2-}$ anions to a ceramic tablet fired at a maximum temperature of the selected range.

When the temperature reaches above 700 °C, the X-ray amorphous CPP/CC sample transforms into a product with phase composition mainly represented by β-$Ca_3(PO_4)_2$ phase with the presence of amount of β-$Ca_2P_2O_7$ phase (~20 wt % for a sample fired at 800 °C). The formation of the β-$Ca_3(PO_4)_2$ phase occurs in accordance with the heterophase reaction (5). Mass spectrometry data indicate that this process takes place in the temperature range of 400–600 °C. The XRD data are consistent with thermal analysis (Figure 3) and mass spectrometry data (Figure 4).

$$CaCO_3 + Ca_2P_2O_7 = Ca_3(PO_4)_2 + CO_2\uparrow \qquad (5)$$

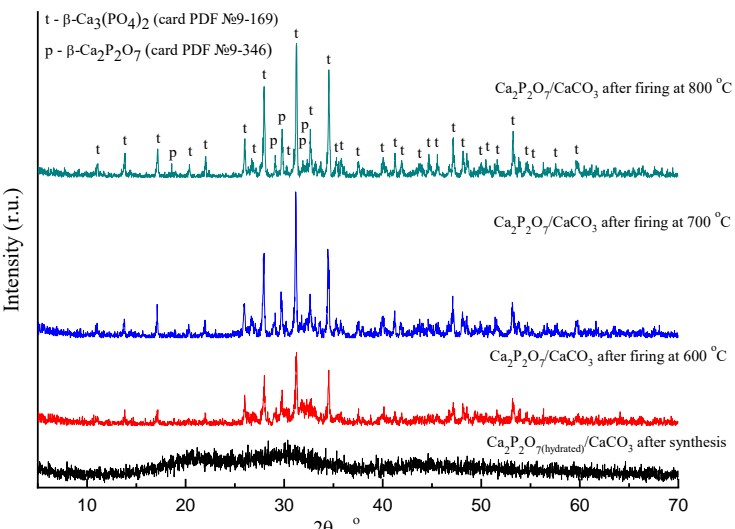

**Figure 2.** XRD data obtained for CPP/CC sample synthesized from a mixed-anionic solution containing $P_2O_7^{4-}$ and $CO_3^{2-}$ after firing at different temperatures: w—$\beta$-$Ca_3(PO_4)_2$ (card PDF №9-169); p—$\beta$-$Ca_2P_2O_7$ (card PDF №9-346).

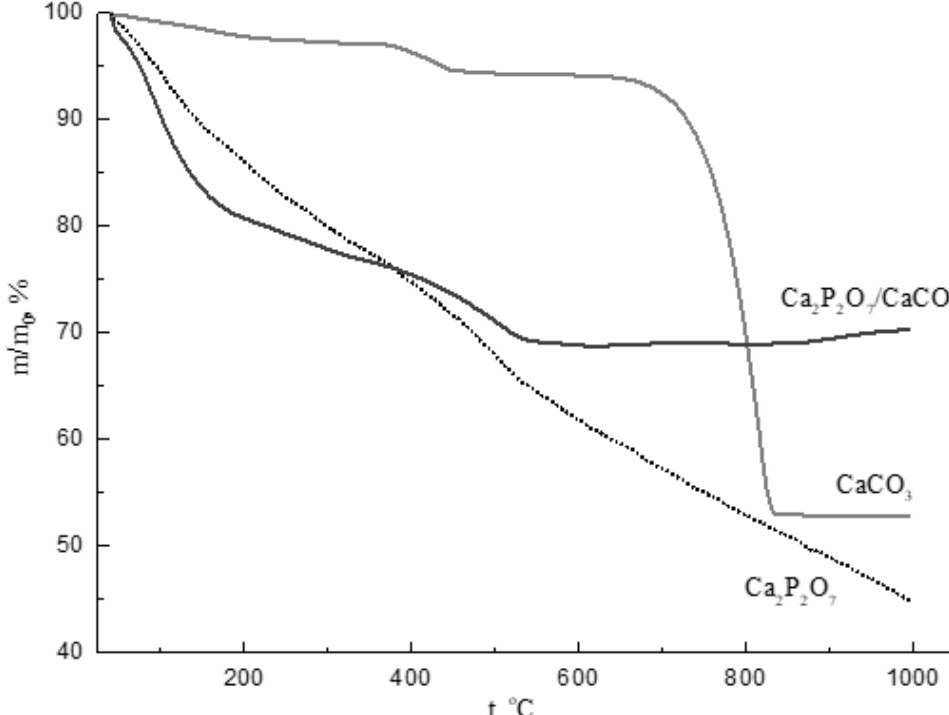

**Figure 3.** Thermal analysis (TG) data for powders synthesized from calcium acetate and ammonium pyrophosphate (CPP-curve); calcium acetate and ammonium carbonate (CC-curve); calcium acetate and mixed-anionic solution containing pyrophosphate and carbonate ions (CPP/CC-curve).

Thermogravimetry curves obtained for the synthesized powders indicate that total mass losses for CC, CPP/CC, and CPP powders were 47, 55, and 30 wt %, respectively. The curve related to the CC sample ($CaCO_3$) reflects the decomposition processes of the by-product ($CH_3COONH_4$) and target product (Equation (6)). CPP-curve ($Ca_2P_2O_7$) reflects the processes of probable by-product decomposition, dehydration of $Ca_2P_2O_7 \cdot xH_2O$, and its conversion to $\beta$-$Ca_2P_2O_7$ in accordance with the reaction (7). The curve for CPP/CC sample ($Ca_2P_2O_7/CaCO_3$) reflects the continuous mass loss associated with both the processes of

$Ca_2P_2O_7·xH_2O$ dehydration and chemical interaction of the charge components resulting in $Ca_3(PO_4)_2$ formation according to the reactions (5) and (8).

$$CaCO_3 = CaO + CO_2\uparrow \tag{6}$$

$$Ca_2P_2O_7·xH_2O = Ca_2P_2O_7 + xH_2O\uparrow \tag{7}$$

$$CaCO_3 + Ca_2P_2O_7·xH_2O = Ca_3(PO_4)_2 + CO_2\uparrow + xH_2O\uparrow \tag{8}$$

A mass loss for a CPP/CC powder synthesized from a mixed-anionic solution reflects the totality of processes occurring in CPP and CC powders.

Indeed, the possibility of interaction between $CaCO_3$ and $Ca_2P_2O_7·xH_2O$ by reaction (8) can be confirmed by the presence of a peak in the range of 150–300 °C of the ion current curve for particles with an m/Z ratio equal to 44, and a peak in the range of 40–200 °C for particles with m/Z ratio equal to 18 for CPP/CC powder. The first stage of mass loss (up to 200 °C) is probably related to the removal of adsorbed water and acetone used as a medium for powder disaggregation after the synthesis and drying. Since the by-product in the syntheses for all samples is $CH_3COONH_4$, the mass spectrometric curves for m/Z 44 contain peaks at 400 °C (CC sample) and 500 °C (CPP and CPP/CC samples). The release of $CO_2$ at these temperatures may be due to the decomposition (combustion in air flux) of the salt. In the range of 700–800 °C, an active release of $CO_2$ associated with $CaCO_3$ decomposition is observed for the CC sample.

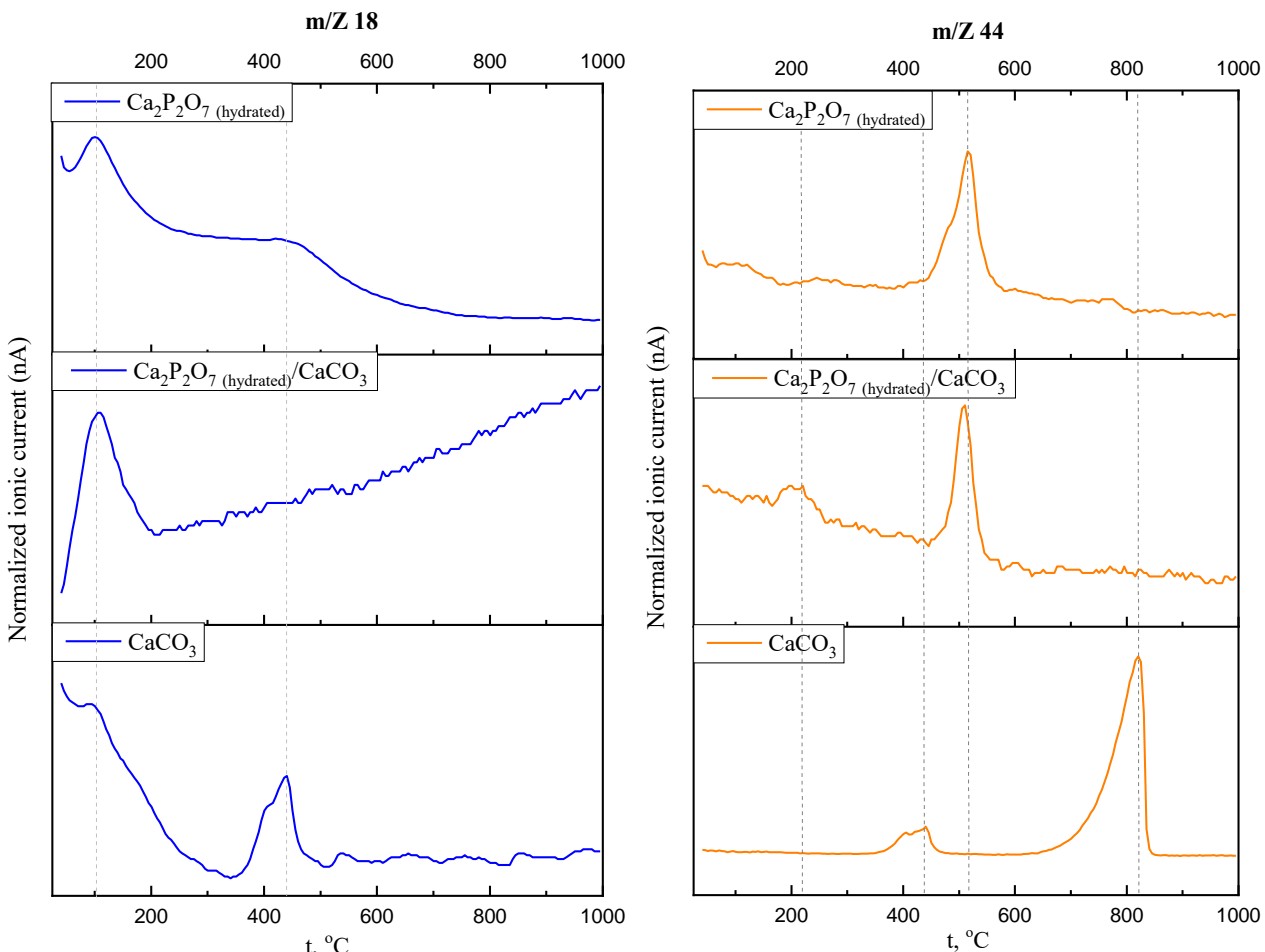

**Figure 4.** Mass spectrometry data: dependences of ion current on temperature for ionized particles with a mass of 18 ($H_2O$) and 44 ($CO_2$) when heating synthesized CPP, CPP/CC, and CC powders.

Figure 5 demonstrates the micrographs of powders synthesized from a mixed-anionic solution after the synthesis and samples after firing at 600, 700, and 800 °C. The particle size after the synthesis is in the range of 100–300 nm. After firing at 600 °C, the particle size does not differ significantly from that of the synthesized ones. After firing at 700 °C, the particle size is in the range of 200–600 nm. Firing at 800 °C leads to a significant grain growth up to 500–2000 nm.

Micrographs of ceramic chips were made for samples of $Ca_2P_2O_7 \cdot xH_2O/CaCO_3$ fired at 700 °C and 800 °C (Figure 6).

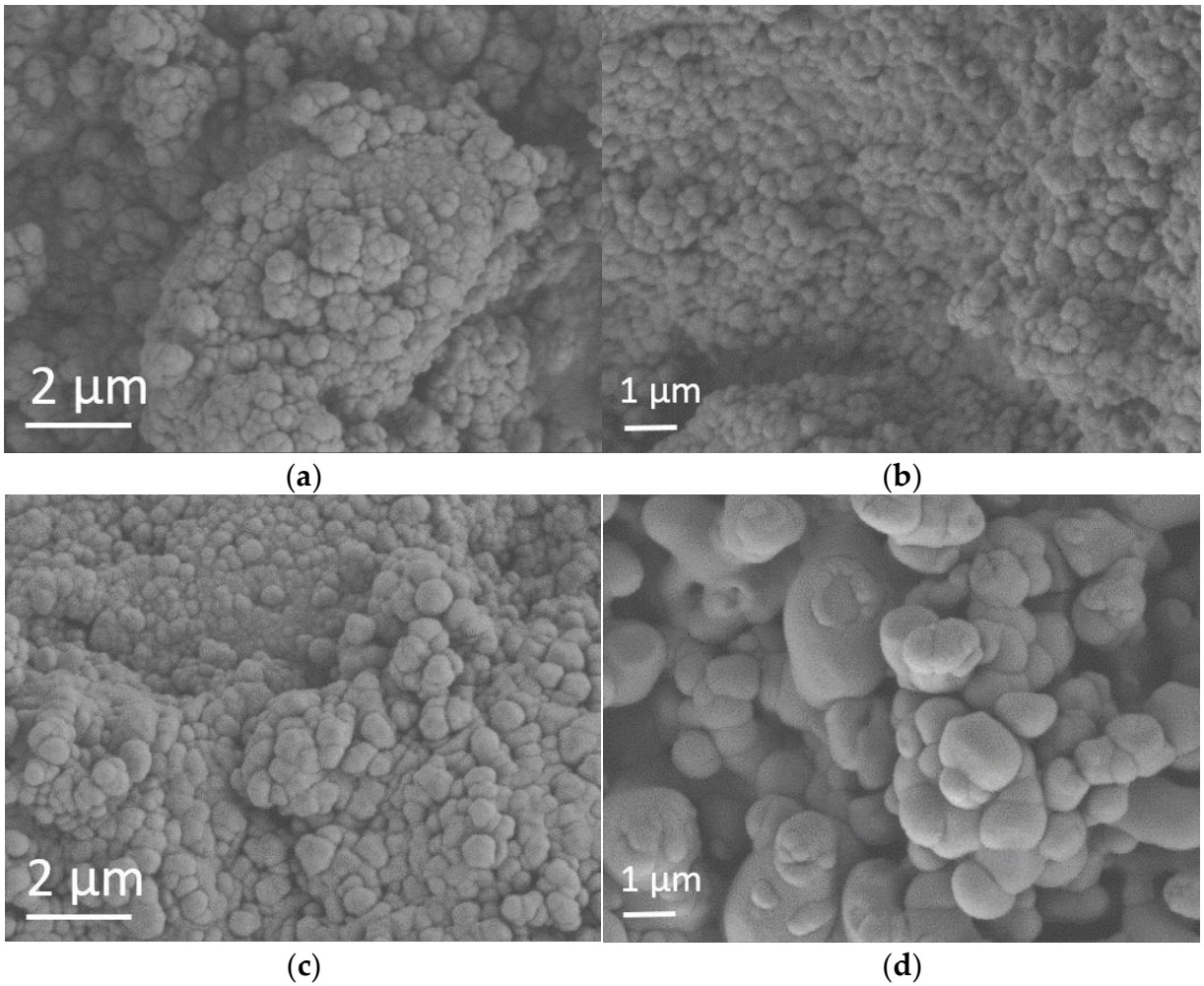

**Figure 5.** Micrographs of CPP/CC powders synthesized from mixed-anionic solution before (**a**) and after firing at 600 °C (**b**), 700 °C (c), and 800 °C (**d**).

Scanning electron microscopy images show that the samples obtained with the introduction of NaCl—$(NaPO_3)_6$ additive contain sufficiently rough aggregates. It should be noted that each aggregate, in turn, consists of smaller particles of a rounded shape with a diameter in the range of 100–200 nm. This fact indicates that the introduction of eutectic composition prevents the growth of particles. This configuration—the presence of sufficiently large aggregates and small particles at the same time—can provide the densest packing of aggregates and, consequently, the greatest density (smaller particles will be located in the space between large ones).

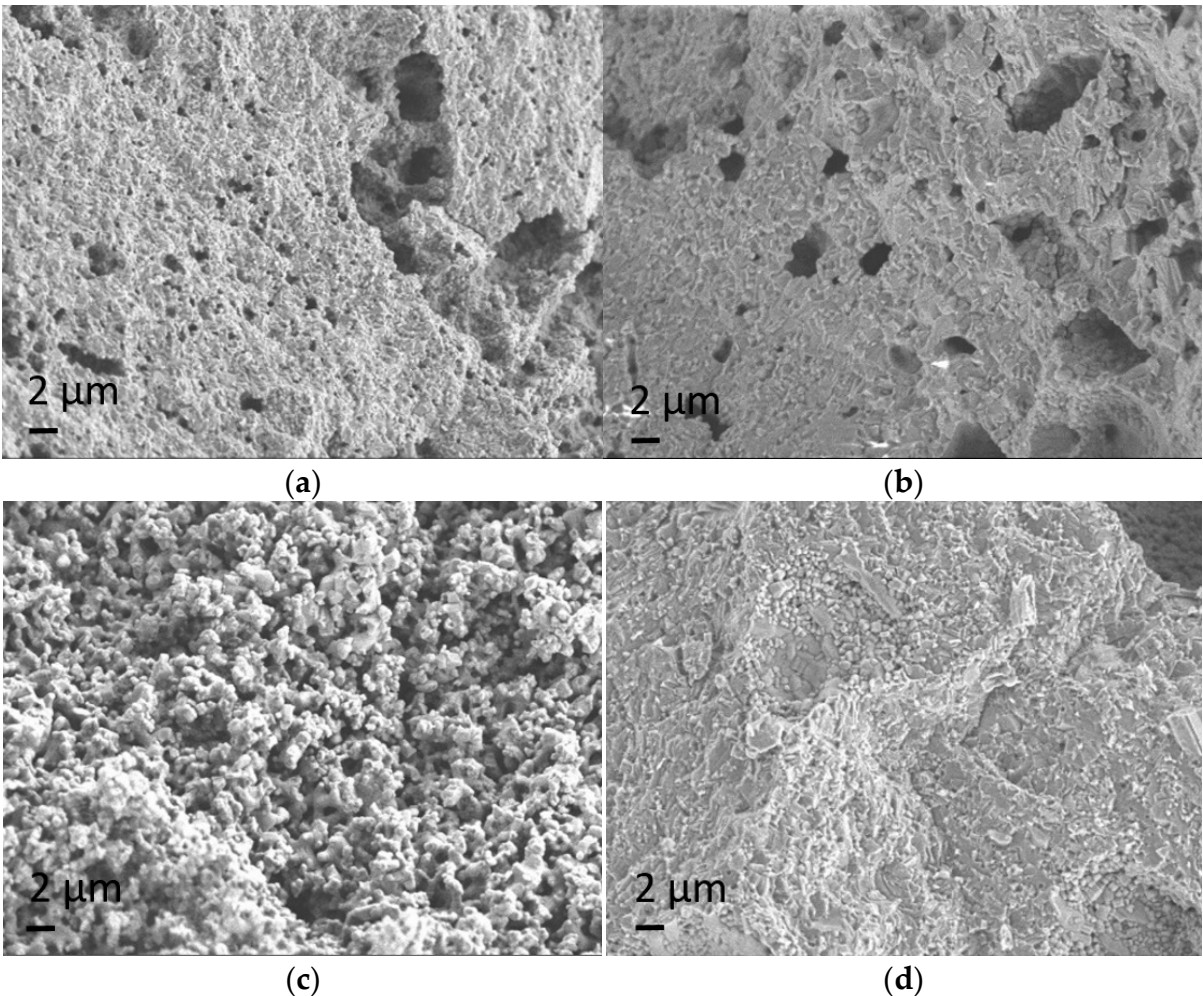

**Figure 6.** Micrographs of the ceramic samples prepared based on the synthesized CPP/CC powder at a pressure value of 100 MPa: (**a**)—chip of a tablet fired at 700 °C without the introduction of an additive; (**b**)—chip of a tablet fired at 800 °C without the introduction of an additive; (**c**)—chip of a tablet fired at 700 °C with the introduction of an additive; (**d**)—chip of a tablet fired at 800 °C with the introduction of an additive.

Additionally, quite large cavities (3–5 μm) are visible on micrographs. These cavities are places of probable crystallization of $CH_3COONH_4$, which burns with the formation of $CO_2$ and $N_2$ gases during the firing. Cavities may affect the properties of samples since the presence of cavities reduces the density and, accordingly, the strength of the ceramics. However, the achieved microstructure and density of ceramic material based on the CPP/CC sample are quite acceptable for the manufacture of biomedical products. For effective integration into bone tissue, such ceramics must have bimodal porosity. Pores with a size of 3–5 μm provide the surface roughness necessary for osteogenic cell attachment and differentiation. Probably, macroporosity (with a pore size of 100–1000 μm) in such material can be formed using additive rapid prototyping technologies.

Figure 7 shows data on changes in shrinkage of the samples with the increasing temperature. The obtained data reflect the packing processes occurring in the cylindrical samples. Based on the measurements of shrinkages, it can be assumed that the addition of the additive NaCl—$(NaPO_3)_6$ has a beneficial effect on the processes of sintering and compaction of the structure. In addition, the additive is biocompatible and is prone to the formation of double salts with the main product, the presence of which leads to the appearance of a liquid phase, accelerating the sintering process.

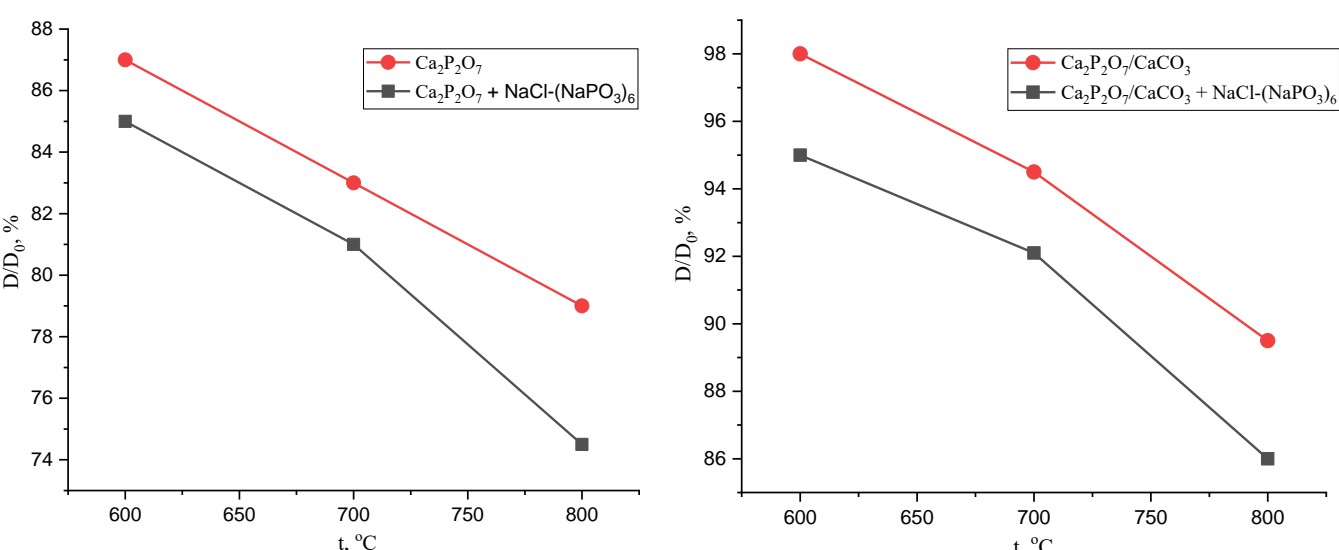

**Figure 7.** Dependences of the relative diameter of the ceramic CPP and CPP/CC samples prepared with or without the use of the additive on the firing temperature.

## 4. Conclusions

Calcium carbonate powder (calcite/vaterite modifications), amorphous calcium pyrophosphate powder, and a powder of mixed anionic composition were synthesized by precipitation from aqueous solutions of calcium acetate, ammonium pyrophosphate, and/or ammonium carbonate. The evolution of phase composition of a product obtained by co-precipitation of $P_2O_7^{4-}$ and $CO_3^{2-}$ anions was discussed based on X-ray diffraction and thermal analysis data. Thus, $\beta$-$Ca_3(PO_4)_2$ and $\beta$-$Ca_2P_2O_7$ phases soluble in aqueous media were observed to be the result after the thermal treatment at 800 °C. The morphology study revealed the simultaneous presence of large aggregates and small particles in the samples, which probably contributes to the formation of a denser packing when fabricating bioceramics. A significant grain enlargement up to 2 μm as a result of sintering along with the special porosity ensured by the presence of micropores with the size of 3–5 μm may have a good effect on the material's mechanical properties and biocompatibility, respectively. Despite the favorable for cell proliferation aspect, the presence of such a porosity requires additional study of micropores effect on the strength and density of carbonate-containing ceramics. An additive of the eutectic composition NaCl—$(NaPO_3)_6$ was proposed, which allows for lowering the firing temperature and achieving a better shrinkage of carbonate-containing ceramics.

**Author Contributions:** Conceptualization, K.P. and T.V.S.; validation, K.P., T.V.S. and G.K.; investigation, K.P., T.V.S., G.K., T.S. and Y.F.; writing—original draft preparation, K.P. and T.V.S.; writing—review and editing, K.P. and J.V.R.; supervision, T.V.S. and J.V.R. All authors have read and agreed to the published version of the manuscript.

**Funding:** This research was funded by the Russian Foundation for Basic Research, grant No. 20-03-00550 A.

**Institutional Review Board Statement:** Not Applicable.

**Informed Consent Statement:** Not Applicable.

**Data Availability Statement:** Not Applicable.

**Acknowledgments:** This research was carried out using the equipment of MSU Shared Research Equipment Center "Technologies for obtaining new nanostructured materials and their complex study" and purchased by MSU in the framework of the Equipment Renovation Program (National Project "Science") and in the framework of the MSU Program of Development.

**Conflicts of Interest:** The authors declare no conflict of interest.

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
