# Peer review of "Powders Based on Ca2P2O7-CaCO3-H2O System as Model Objects for the Development of Bioceramics"

_ceramics, doi:10.3390/ceramics5030032_

Round 1

Reviewer 1 Report

Overall, this is a good proposal. However, should need to address a few issues for further consideration. 

General comments:

There are already similar works reported by the same authors [Ref below]. Therefore, should explain the importance, novelty and significance of this study compared to earlier works.

Article Title: Bioceramics Based on β-Calcium Pyrophosphate

Article Title: Ca2P2O7–Ca(PO3)2 Ceramic Obtained by Firing β-Tricalcium Phosphate and Monocalcium Phosphate Monohydrate Based Cement Stone.

The data presented in this study are not enough to claim the conclusion. Try to add some more data, which are accessible at the Authors´ institute.

Specific comments:

At the end of Intro:  Thus, the development of ceramic materials.: This is not a solid or convincing statement to support the significance of this study.

2.1. Synthesis of Powders: Be specific here, about what you are fabricating

Figure 1: Try to reduce the noise peak

Figure 2: Label the axis. w - β-Са3(РО4)2 (card PDF â„–9-169); Ñ€ - β-Ca2P2O7: Not provided in the image. 

Figure 3: Provide the sample information in the legend

Figure 4:Same as Fig.3

Figure 5: Explain the sample details in the legend

- β-Са3(РО4)2 & - β-Ca2P2O7: Provide the details of this sample in the method section.

The discussion section is very poor, Explain the reason behind the changes in the study with earlier references and discuss the possible hypothesis of each parameter. Also, try to discuss the earlier findings (Papers) reported by the Authors and compare the present study. 

Author Response

Dear Sir/Madam,

Thank you for your valuable comments on our research.

We added several changes to the manuscript in accordance with your recommendations.

1) On the page 2, you may find our brief explanation of importance and novelty of the work. In general, our group tries to develop the concept of composite ceramics based on calcium phosphates and some other biocompatible components, such as calcium carbonate. In our earlier research, we paid attention to calcium orthophosphate in combination with carbonate. In this study, we decided to consider pyrophosphate instead, due to the reasons described in the revised manuscript. There are definitely a significant number of studies devoted to calcium pyrophosphate, but the combination of pyro-/carbo is poorly presented.

The studies you have given as an example demonstrate other approaches of the synthesis (mechanical activation) and observe calcium pyrophosphate as an individual component, as a rule. In addition, in our research we tried to apply not ordinary method of the synthesis using ion exchange.

2)We have changed the Conclusions section and improved the data by adding thermal analysis data (thermogravimetry and mass spectrometric detection).

3) We have improved the Figures in accordance with your recommendations (please see the attached file).

4)We have added several discussions on materials properties, including phase evolution and thermal behavior, to the Results and Discussions section, so it can be filled with more specific information required for the study. However, the comparison was mainly conducted with our earlier works. 

We are ready to add more comparisons and references to the related research works if you deem it necessary. 

Thank you again for your significant comments, your time, and consideration.

Best regards,

Kristina Peranidze

Reviewer 2 Report

Please state the novelty of the paper. The performed chemical transformations are commonly known, and the phase transitions when thermal treating the samples is not innovative.

It is hard to understand why the authors choose to synthesize CaCO3 and H3PO4, rather than use commercial ones, with a high purity degree, especially since, as stated, the processes may not proceed completely or generated secondary phases.

Please rephrase: “With increasing temperature, x-ray amorphous powder transforms into a powder”.

The results are merely presented, and a proper Discussion section is missing. 

Author Response

Dear Sir/Madam,

Thank you for your valuable comments on our research.

We tried to add certain changes to the manuscript in accordance with you recommendations.

1) You may find a brief explanation of the novelty on the page 2 of the revised manuscript. The main idea of the research is the attempt to develop composite ceramics containing calcium phosphate and another biocompatible component, such as calcium carbonate. In our earlier study, we proposed to consider orthophosphate-carbonate system. Here, we are discussing calcium pyrophosphate in combination with carbonate, which are poorly presented in the literature. We hope to find ways of CaCO3 introduction to the ceramics and use possible positive effects of it. Considering the specificity of both CaCO3 and Ca2P2O7 described in the Introduction Section, it seems to be a challenging and interesting task.

2) In our research, we prefer to adhere to natural approaches, in which the material with all the features of phase composition, morphology etc. is formed directly during the synthesis. We did use non-ordinary synthesis method in particular to study chemical and physical properties of the powders. In addition, this method that includes the ion exchange have not been presented to a significant extent.

3) We have rebuilt the phrase "With increasing temperature..." (page 6).

4) We have also improved the section Results and Discussions (please find attached the revised manuscript), but have not separated Discussions from the Results, since we find it inconvenient.

Thank you for your comments, time, and consideration.

Best regards,

Kristina Peranidze

Reviewer 3 Report

Interesting and correctly performed work on the use of Ca2P2O7-CaCO3 System

Only a few criticisms to report:

- In the final section of the abstract, insert a sentence on the possible clinical implications of the research carried out in the study

- check that all keywords are Pubmed MESH terms

- At the end of the introduction section, the null hypotheses of the study must be inserted and must be refuted in the light of the results obtained in the study

- Some considerations are missing in the introduction section on the role that bioactive materials are having in medicine in general and in dentistry in particular. In this regard, I recommend that you insert the following scientific work in the reference section, which could be of help to the reader

Lardani L, Derchi G, Marchio V, Carli E. One-Year Clinical Performance of Activa ™ Bioactive-Restorative Composite in Primary Molars. Children (Basel). 2022; 9 (3): 433. Published 2022 Mar 19. doi: 10.3390 / children9030433

Author Response

Dear Sir/Madam,

Thank you for your valuable comments.

We tried to add certain changes to the manuscript.

1) We added several sentences on possible clinical application of the ceramic materials in the system under discussion to Introduction Section (instead of Abstract). Please find attached the revised manuscript (page 2).

2) We decided not to add null hypothesis of the research, since the concept, as we assume, is rather simple: we compose a material between two control points - Ca2P2O7 and CaCO3. The ultimate material obtained by co-precipitation is assumed to possess the positive features of both components.

3) As you recommended, we added a research on bioactive dental material that you suggested.

Thank you for your time and consideration.

Best regards,

Kristina Peranidze

Round 2

Reviewer 1 Report

The revised file is satisfactory and addressed all my comments. 

Reviewer 2 Report

The authors addressed my comments.